# An Entropy-Based Directed Random Walk for Cancer Classification Using Gene Expression Data Based on Bi-Random Walk on Two Separated Networks

**DOI:** 10.3390/genes14030574

**Published:** 2023-02-24

**Authors:** Xin Hui Tay, Shahreen Kasim, Tole Sutikno, Mohd Farhan Md Fudzee, Rohayanti Hassan, Emelia Akashah Patah Akhir, Norshakirah Aziz, Choon Sen Seah

**Affiliations:** 1Faculty of Computer Sciences and Information Technology, Universiti Tun Hussein Onn Malaysia, Batu Pahat 83000, Malaysia; 2Department of Electrical Engineering, Universitas Ahmad Dahlan, Yogyakarta 55166, Indonesia; 3School of Computing, Faculty of Electrical Engineering, Universiti Teknologi Malaysia, Skudai 81310, Malaysia; 4Department of Computer and Information Sciences, Universiti Teknologi Petronas, Seri Iskandar 32610, Malaysia; 5Faculty of Accounting & Management, Universiti Tunku Abdul Rahman, Kajang 43000, Malaysia

**Keywords:** directed random walk, pathway-based analysis, cancer classification

## Abstract

The integration of microarray technologies and machine learning methods has become popular in predicting the pathological condition of diseases and discovering risk genes. Traditional microarray analysis considers pathways as a simple gene set, treating all genes in the pathway identically while ignoring the pathway network’s structure information. This study proposed an entropy-based directed random walk (e-DRW) method to infer pathway activities. Two enhancements from the conventional DRW were conducted, which are (1) to increase the coverage of human pathway information by constructing two inputting networks for pathway activity inference, and (2) to enhance the gene-weighting method in DRW by incorporating correlation coefficient values and *t*-test statistic scores. To test the objectives, gene expression datasets were used as input datasets while the pathway datasets were used as reference datasets to build two directed graphs. The within-dataset experiments indicated that e-DRW method demonstrated robust and superior performance in terms of classification accuracy and robustness of the predicted risk-active pathways compared to the other methods. In conclusion, the results revealed that e-DRW not only improved the prediction performance, but also effectively extracted topologically important pathways and genes that were specifically related to the corresponding cancer types.

## 1. Introduction

The accurate prediction of prognosis and the metastatic potential of cancer is a major challenge in clinical cancer research. Through the evolution of high-throughput technologies, deoxyribonucleic acid (DNA) microarray analysis can classify tumor samples overriding the traditional diagnostic methods. This technology allows for the extraction of a huge amount of molecular information, which aids in the discovery of tumor-specific biomarkers. However, the reproducibility of individual gene biomarkers has been challenging, as the identified gene markers in one dataset failed to predict the same disease phenotype obtained in other datasets [1]. This discrepancy is usually due to the cellular heterogeneity within tissues, the inherent genetic heterogeneity across patients, and the measurement error in microarray platforms [2]. Furthermore, microarray analysis of gene expression data generally produces plenty of genes from patients with the same diseases, hence, leading to a high dimension small sample size problem. All of these factors often decrease the prediction performance and reproducibility of individual gene biomarkers in independent cohorts of patients.

To address the unreliable or inconsistent prediction of gene biomarkers in datasets, biological pathway data were introduced to identify robust pathway biomarkers in functional categories [3,4,5,6]. As gene products are known to function coordinately in functional modules, the mutual interest between the pathway data and gene expression data can extract function-related genes to produce consistent and reproducible biomarkers [7]. Such biomarkers at the functional level can reduce the impact of noise in the microarray data by allowing for a more accurate biological interpretation of the disease-canonical pathway correlations [2]. In fact, several studies [3,4,8,9] have shown that pathway markers are more reliable compared to single gene markers as they provide crucial biological insights into the underlying processes that give rise to various disease phenotypes. Furthermore, pathway-based classifiers often achieve comparable or better classification performance compared to conventional gene-based classifiers [4,5].

In cancer classification, a robust gene weight merit is vital to reflect the importance of genes from different aspects and establish significant genes with related diseases [10]. Several studies in pathway-based methods typically use the *t*-test as the gene weighting method to measure the gene expression levels for further cancer classification. Consider that those existing pathway-based methods including directed random walk (DRW), significant DRW, and pathway activity inference using condition-responsive genes (PAC method) all targeted on the *t*-test as the single statistical measurement to weigh each gene in the gene expression data. However, the lack of a comprehensive gene weighting method could affect the classification performance of pathway-based methods [10,11].

To combat the aforementioned issue, this study proposed an entropy-based directed random walk (e-DRW) on two separated biological networks that enhances the accuracy of cancer classification. Two inputting networks were proposed for the random walking of e-DRW, which comprises 328 KEGG pathways collected from the KEGG database [12] and 208 pathways gathered from the Pathway Interaction Database (PID) [13]. The representation of two biological networks (KEGG-PID) is known as the directed pathway network. An improved gene weighting strategy using point biserial correlation (PBC) coefficients and the T-test was proposed in e-DRW. The gene weighting method modeled the combined effect of the statistical measurement of the gene expression levels with the class label (normal, cancer). The weight initialization of genes and the scoring of pathways were further enhanced by the application of the entropy metric to calculate the pathway activity score. The proposed method was implemented in the R platform with version 4.2.1 in 64-bit using Windows 10 (refer the supplementary e-DRW R package for more details). Figure 1 shows the workflow of e-DRW. The steps involved in e-DRW include data pre-processing and the construction of biological networks, normalization based on z-scores, differential expression analysis, entropy-based directed random walk, entropy-based pathway activity inference, and classification.

## 2. Materials and Methods

This section presents the materials and methodology used in e-DRW. Based on Figure 1, each of the steps involved in the workflow of e-DRW will be described thoroughly.

### 2.1. Data Pre-Processing and Construction of Biological Networks

The first step in e-DRW is data pre-processing and the construction of biological networks. Six gene expression datasets were obtained from the National Center for Biotechnology Information (NCBI) Gene Expression Omnibus (GEO) database, which are lung [14], stomach [15], liver [16], kidney [17], thyroid [18], and breast [19] cancer datasets. The collected datasets undergo data pre-processing to produce the cleaned gene expression data. There are two phases involved in data pre-processing: (i) data cleaning and imputation, and (ii) the normalization of the gene expression data. In the first phase, the unwanted and empty values of the attributes were removed. The unwanted attributes include patient biological information and dataset information that is not applicable in cancer classification whereas empty values of attributes refer to the missing values that appeared across the rows in the gene expression dataset. Then, rows with incomplete values of attributes were imputed with mean values to resolve the inconsistencies in data. The completed dataset following the application of the mean imputation was used for inference. However, the rearrangement of data was run through before proceeding to the next phase. The normalization step in the second phase typically included thresholds or flooring to remove poorly detected probes and log2 transformation to normalize the distribution of probes across the intensity range of the experiment. Gene Pattern was used for dataset pre-processing to remove platform noise and genes that have little variation [20]. Table 1 shows the details of the datasets after data preprocessing.

On the other hand, a directed pathway network was constructed based on the pathway information obtained from the KEGG database and PID database. First, each KEGG pathway was converted into a directed graph using the NetPathMiner [21] software package. A total of 328 human pathways were merged to form the KEGG network, covering 6667 nodes and 116,773 directed edges. Subsequently, each PID pathway was converted into another directed graph using the PaxtoolsR [22] software package. A total of 208 human pathways were merged to form the PID network, covering 2817 nodes and 39,289 directed edges. Each node in the graph represented a gene, while each directed edge represented how the genes interacted and controlled each other. The direction of the edge was determined by the type of interaction between the two genes found in the KEGG and PID pathway databases.

### 2.2. Normalization Based on Z-Scores

The second step in e-DRW is normalization based on z-scores. The collected gene expression datasets underwent normalization based on the z-score to produce the normalized gene expression data. This step aimed to normalize the gene expression values over all samples to a scale of mean zero and variance one [2,23]. Normalization based on z-scores can provide a way to standardize data across the gene expression dataset. This is an important step to achieving good classification performance before evaluating the data on machine learning algorithms [24]. The formula of normalization based on z-scores is shown as below:(1)$$z\left(gi\right)=\frac{gene\left(gi\right)-\bar{X}\left(gi\right)}{S\left(gi\right)}\;$$
where *z*(*gi*) is the normalized gene expression values for gene *i* over all samples; *gene*(*gi*) is the gene expression values for gene *i* over all samples; $\bar{X}$(*gi*) is the mean of gene expression values for gene *i*; *S*(*gi*) is the standard deviation of the gene expression values for gene *i*; and *i* is the number of genes in the gene expression data.

### 2.3. Differential Expression Analysis

The third step in e-DRW is differential expression analysis. *t*-test statistics with equal variances [25] and the point biserial correlation (PBC) coefficient [26] were calculated for each gene in the gene expression data. This step aimed to calculate the statistical difference between the normal and disease samples. The formula for calculating the *t*-test statistics with equal variances of each gene is shown below:(2)$$t\left(gi\right)=\frac{{\bar{X}}_{1}-{\bar{X}}_{2}}{S\times \sqrt{\left(\frac{1}{N_{1}}\right)+\left(\frac{1}{N_{2}}\right)}}\;$$
where *t*(*gi*) is the t-score of the *t*-test statistics with equal variances, $\bar{X}$_1_ is the mean for the normal sample; $\bar{X}$_2_ is the mean for the tumor sample; *S* is the standard deviation of the two samples; *N*_1_ is the number of normal samples; and *N*_2_ is the number of tumor samples. At the same time, PBC was performed to calculate the correlation coefficient for each gene. PBC measures the relationship between two variables (genes) using the formula shown below:(3)$$p_{pb}\left(gi\right)=\frac{M_{1}-M_{2}}{S}\sqrt{pq}\;$$
where *p**_pb_*(*gi*) is the PBC coefficient for each gene *gi*; *M*_1_ is the mean for normal samples; *M*_2_ is the mean for the tumor sample, *S* is the standard deviation of the normal and tumor samples; *p* is the proportion of cases in normal samples; and *q* is the proportion of cases in the tumor samples.

This study employed a combination of PBC and *t*-test scores (PCT scores) as a gene-weighting method. The weighted expressions of the member genes reflected two factors: (1) the degree of the differential expression of genes between the means of the normal and cancer group; and (2) the correlation between a gene expression and class label (normal, cancer). Based on these considerations, a new robust gene-weighting method was proposed in this study. The normalized expression values of gene *gi* in sample *k* are defined as:*Z(gi) = t(gi)^2^ + |p(gi)|*(4)
where *t*(*gi*) is the t-score of gene *gi* calculated using a two-tailed *t*-test between two phenotypes, while *ρ*(*gi*) is the absolute PBC between gene *gi* and the class label. *Z*(*gi*) represents the weighted normalized expression (PCT scores) of gene *gi* in sample *k*, reflecting the differential expression degree of gene *gi* and its correlation with the phenotype. Larger expression values *Z*(*gi*) can be related to higher differential expression and a larger correlation with the phenotype.

### 2.4. Entropy-Based Directed Random Walk (e-DRW)

The fourth step in e-DRW is the calculation of the genes’ weight in the directed graph. Before implementing e-DRW, the initial weight of the genes was first calculated using the formula shown below:(5)$$Wo=\frac{absolute\left(Z\left(gi\right)\right)-maximum\left(Z\left(gi\right)\right)}{maximum\left(Z\left(gi\right)\right)-minimum\left(Z\left(gi\right)\right)}\;$$
where *Wo* is the initial weight of genes; *absolute*(*Z*(*gi*)) is the absolute values of PCT score; *maximum*(*Z*(*gi*)) is the maximum values of PCT score; and *minimum*(*Z(gi*)) is the minimum values of the PCT score. Then, the entropy [27] of each gene was used as the weight parameter to calculate the distribution of each node in the directed graph. Furthermore, the directed graphs for the KEGG and PID networks were converted to an entropy edge-weighted adjacency matrix (network entropy) to enhance the calculation of the genes’ weight in both networks. The calculated node entropy for each gene, KEGG, and PID network entropy were then implemented in e-DRW for the pathway activity inference. e-DRW is defined as:(6)$$H_{t+1}=\left(1–r\right)E^{T}H_{t}+rH_{0}\;$$
where *H_t_* represents the node entropy vector that holds the probability at the specific node at time step *t*. *H*_0_ is the initial entropy probability vector; *E^T^* is an entropy edge-weighted adjacency matrix developed from the directed graphs (with edges); *r* denotes the restart probability ranges from 0.1 to 0.9; and *H_t_*_+1_ denotes the final entropy probability vector.

Considering the bi-random walk of e-DRW on two inputting networks (KEGG and PID networks), the random walking of e-DRW was implemented on the two networks successively to obtain the separate results. The random walk processes are illustrated by the following equations:(7)$$\mathrm{KEGG}\;\mathrm{network}:\;H^{G}{}_{t+1}=\left(1–r\right)G^{E}H_{t}+rH_{0}$$
(8)$$\mathrm{PID}\;\mathrm{network}:\;H^{P}{}_{t+1}=\left(1–r\right)P^{E}H_{t}+rH_{0}$$
where *G^E^* represents the entropy edge-weighted adjacency matrix of the KEGG network, and *P^E^* represents the entropy edge-weighted adjacency matrix of the PID network. The separate results were then applied for further pathway activity inference and cancer classifications.

### 2.5. Entropy-Based Pathway Activity Inference

The fifth step in e-DRW is entropy-based pathway activity inference. The normalized gene expression data were first split into three subsets whereby 60% of the datasets was used as the training set, 20% used as the validation sets, and another 20% used as the test sets. The three subsets were then utilized for entropy-based pathway activity inference. Genes with *p*-values less than 0.05 for each pathway in the pathway data were chosen to construct the pathway activities. Entropy-based pathway activity inference for the training, validation, and test sets for the KEGG and PID networks is defined as:(9)$$a\left(Pj\right)=\frac{\sum_{i=1}^{nj}H\infty \left(gi\right)\times PCTscore\left(gi\right)\times Z\left(gi\right)}{\sqrt{\sum_{i=1}^{nj}(H\infty \left(\frac{1-gi}{sum\left(1-gi\right)}\right))²}}\;$$
where *a*(*Pj*) is the pathway activity (or expression value vector); *H∞* is the output of genes (or weight vector calculated from e-DRW); *PCTscore*(*gi*) is the summation of PBC between gene *gi* and class label (normal and tumor samples) and the *t*-test statistics of gene *gi* from a two-tailed *t*-test with equal variances in the expression values between two classes. *Z*(*gi*) is a normalized value vector of gene *gi* across the whole dataset, and $H\infty \left(\frac{1-gi}{sum\left(1-gi\right)}\right)$ is the entropy weight of gene *gi*. The calculated KEGG and PID pathway expression profiles for the training, validation, and test sets were then combined respectively for the pathway selections. The top 50 pathways ranked by the *t*-test statistics for the training, validation, and test sets were selected to construct the final pathway expression profiles for further classification.

### 2.6. Classification

The final step in e-DRW is classification. Within-dataset experiments were implemented for the six cancer datasets. The R caret [28] package was utilized to obtain the classification accuracy. Three classifiers were selected to evaluate the performance of e-DRW, which were Naïve Bayes (NB), K-nearest neighbors (KNN), and logistic regression (GLM). e-DRW implemented stratified 10-fold cross validation on the training set to evaluate the performance of the classifier. The top 50 pathways in the training dataset were used as candidate features to build the model. Subsequently, pathways were added sequentially to train the model. The performance of the classifier was measured by evaluating the area under the receiver operating characteristic curve (AUC). The added pathway marker was maintained in the feature set if the AUC increased, but was removed if otherwise [2]. This process was repeated for the top 50 pathway markers to optimize the classifier and to yield the best feature set. The performance of the optimized classifier was evaluated on the test set using pathway markers from the best feature set. This process was repeated 10 times to ensure unbiased evaluation and to estimate the variation of the AUC. As the final step, the mean AUC across 10 classifiers was estimated to represent the overall performance of the classification method.

## 3. Results

This section presents the classification performance within-dataset experiments. For comparison with other pathway activity inference methods, five pathway-based classification methods were chosen, namely, the DRW method [2], sDRW method [29], iDRW method [30], PAC method [4], and principal component analysis (PCA method) [7]. The experimental setting was the same for the DRW, sDRW, iDRW, and PAC methods. The PCA method was implemented as the pathway-based classification method by applying the proposed KEGG network to calculate the pathway expression profiles. Classification accuracy and robustness of the predicted risk-active pathways were chosen as the performance measurements of cancer classification.

### 3.1. Classification Performance on Within-Dataset Experiments

Table 2 presents the mean AUCs of e-DRW with varying restart probabilities (0.1–0.9) across the six cancer datasets using three different classifiers (NB, KNN, LR).

Based on Table 3, the KNN classifier showed the highest mean AUCs across four cancer datasets, which were the lung cancer dataset, stomach cancer dataset, liver cancer dataset, and breast cancer dataset. Hence, the KNN classifier was chosen to evaluate the classification performance of the other methods. For a fair and effective comparison with other methods, within-dataset experiments similar to those used in Liu et al. (2013) [2] were implemented to evaluate the classification performance. Figure 2 illustrates the comparison of the classification performance for the six methods on the within-dataset experiments.

Based on Figure 2, the proposed e-DRW method obtained mean AUCs of 0.980374 for the lung cancer dataset, 0.955072 for the stomach cancer dataset, 0.931034 for the liver cancer dataset, 0.906522 for the kidney cancer dataset, 0.954286 for the thyroid cancer dataset, and 0.769068 for the breast cancer dataset. By comparing the mean AUCs with other methods, e-DRW achieved the highest mean AUCs across all datasets, except for the lung cancer dataset, which was slightly lower than the iDRW (0.981259) method. This indicates that e-DRW-based pathway markers are quite competent in discriminating between different disease phenotypes. It also demonstrated the best overall classification performance on the within-dataset experiments when compared with other methods.

### 3.2. Robustness of Predicted Risk-Active Pathways

The detection of robust risk-active pathways is important in cancer studies. Risk-active pathways detected across 10 experiments for each cancer dataset are provided in Appendix A. Genes in the risk-active pathways were extracted and provided in Appendix A. Table 3 lists the top 15 most predicted cancer-related pathways involved in various biological processes studied by e-DRW across the six datasets.

Based on Table 3, pathways specific to the phenotype of classification were identified. Among these pathways, the PI3K-Akt signaling pathway and pathways in cancer identified in most cancer datasets reported the relations of these pathways with cancer [2,31,32,33]. Furthermore, the human papillomavirus infection pathway and calcium signaling pathway are known cancer pathways, as reported in multiple studies [34,35,36,37,38,39]. The ECM–receptor interaction pathway and focal adhesion pathway also suggest their important roles in lung and thyroid cancer based on pertinent studies [40,41]. In addition, several extensively researched cancer-related pathways were identified as risk-active pathways in multiple cancers such as the lipid and atherosclerosis pathway, Apelin signaling pathway, Hippo signaling pathway, and Wnt signaling pathway [42,43,44,45,46,47,48,49]. Furthermore, the predictions of the small cell lung cancer pathway, neuroactive ligand–receptor interaction pathway, and integrin-linked kinase signaling pathway were consistent with several studies [50,51,52,53,54,55]. Relevant studies have validated the coactive effect of adrenergic signaling in the cardiomyocyte pathway and the cGMP-PKG signaling pathway with cancers [56,57,58].

## 4. Discussion

In the literature, multiple existing pathway-based methods incorporate pathway topological information to identify important genes within pathways. For instance, Guo et al. [3] employed the mean or median expression value of the member genes to infer the pathway activity. Bild et al. [7] used the first principal component of the expression profile of member genes to evaluate the activity of a given pathway (PCA method). Lee et al. [4] proposed pathway activity inference using only a subset of genes in the pathway, called the condition responsive genes (CORGs), in which the combined expression levels can accurately discriminate the phenotypes of interest (PAC method). However, these methods simply consider pathways as simple gene sets but ignore significant individual genes and interactions between genes, which are essential to infer a more robust pathway activity [1].

A comprehensive pathway topology is important to clarify the roles that the genes play in the pathway and weight the genes more precisely [2]. Several pathway-based methods utilize pathway topology information collected from pathway databases for analysis. For example, Liu et al. [2] constructed the global-directed pathway network, which covers 300 pathways collected from the Kyoto Encyclopedia of Genes and Genomes (KEGG) pathway database. Seah et al. [29] built a directed graph using 300 pathway datasets obtained from the KEGG pathway database. Kim et al. [30] selected 327 human pathways to construct a directed gene–gene graph for pathway activity inference. Lee et al. [4] collected 472 canonical metabolic and signaling pathways from MsigDB v1.0 for cancer classification. However, the limitations of these methods lie in the coverage of human pathway information. The complete biological pathway information not only enables a more accurate prediction of disease status, but also paves the way to unveiling novel functional pathways or complexes [59,60,61,62].

This study proposed an entropy-based pathway activity inference scheme to identify reproducible pathway biomarkers for clinical cancer applications. Previous literature has revealed that individual gene markers are less reliable compared to pathway markers, and thus are unable to effectively capture the biological interpretation of gene expression in functional categories [3,4,5,6]. The proposed entropy-based pathway activity inference method conducted a bi-random walk of e-DRW on two separated networks for pathway activity inference. A robust gene-weighting method was proposed that incorporates PBC and the *t*-test to calculate the weight of each gene. Considering the effectiveness of entropy as weight variables, entropy was implemented as a weight parameter to enhance the weight initialization scheme in e-DRW [63]. The entropy weight metric was also applied in entropy-based pathway activity inference to enhance e-DRW pathway activities for cancer classification.

Based on the classification performance, the mean AUCs of the e-DRW method were significantly higher and more robust across the experiments. The reliable performance of the e-DRW pathway activities could be attributed to the construction of the directed pathway network and gene-weighting method. The proposed biological networks provide larger pathway topology for the random walking of e-DRW on the KEGG and PID networks. Furthermore, the gene-weighting method based on PBC and the *t*-test can greatly magnify the signals of essential genes whose expression levels may have a large impact on the pathway while weakening the differential expression of genes that only appear downstream or have a minor impact on the system. Therefore, the e-DRW approach could alleviate the noise caused by sample heterogeneity or technical measurements, resulting in more reproducible pathway activities.

Moreover, the mean AUCs of e-DRW were better in terms of cancer classification due to higher accuracy compared to other pathway-based analysis methods. Results on the top 15 known cancer-related pathways showed that the performance of most pathways was very close to the best performance. This indicates that the proposed e-DRW was even robust on many cancer-related pathways. Additionally, we found that the proposed e-DRW could achieve a satisfactory performance for all datasets through the PI3K-Akt signaling pathway and pathways in cancer. Overall, e-DRW was more effective in pathways and gene prediction as it was more robust compared to the other methods.

## 5. Conclusions

In cancer studies, an accurate prediction of cancer is crucial for the diagnosis and prognosis of clinical therapy. An e-DRW on two separated networks for cancer classification was proposed. The two enhancements based on Liu et al.’s work [2] and the proposed e-DRW were proven to be effective in inferring pathway activities and accurate cancer classification. The proposed enhancements included (1) the construction of the directed pathway network (KEGG and PID networks), and (2) gene-weighting based on the PBC and *t*-test. Two biological networks (KEGG and PID networks) were constructed to increase the coverage of human pathway information. A gene-weighting method in e-DRW incorporating the *t*-test statistics scores and correlation coefficient values to weigh each gene in the directed pathway network was also proposed. This weighting strategy not only reflects the degree of the differential expression of genes between the normal and cancer groups, but also considers the correlation coefficient values between genes in the gene expression data. Additionally, the weight initialization of genes and the scoring of pathways were further enhanced by the calculation of gene expression entropy, which implicitly increased the accuracy of cancer classification. Finally, stratified 10-fold cross-validation was utilized to train the classifier and classify the significant pathways detected by e-DRW. In conclusion, the proposed approach was more effective and feasible for cancer classification compared to other pathway-based methods.

## Figures and Tables

**Figure 1 genes-14-00574-f001:**
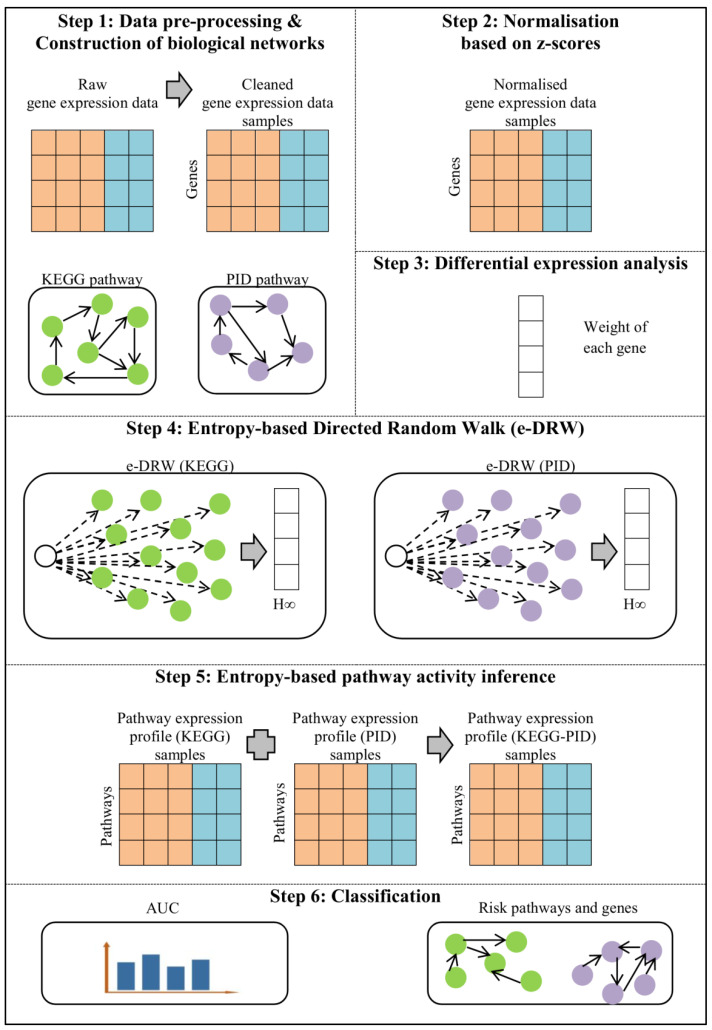
Workflow of e-DRW.

**Figure 2 genes-14-00574-f002:**
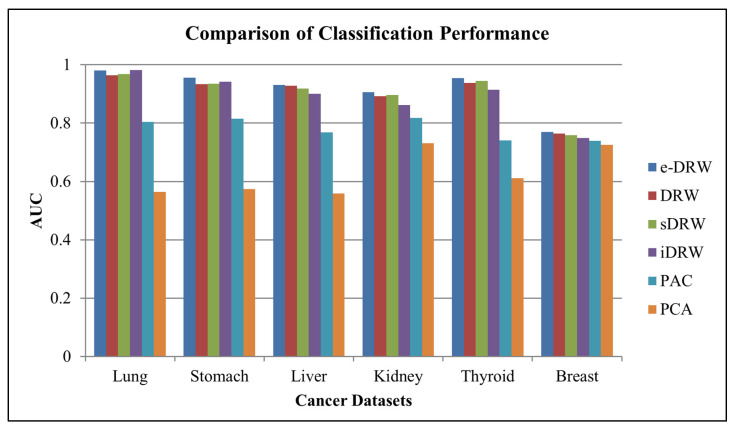
Comparison of classification performance.

**Table 1 genes-14-00574-t001:** Gene expression datasets after pre-processing.

Cancer	GEO ID	Platform ID	Number of Genes		Number of Cancerous Samples	Number of Normal Samples	Total Number of Samples
			Raw	Cleaned			
Lung	GSE10072	GPL96	22283	12986	58	49	107
Stomach	GSE13911	GPL570	54675	12419	38	31	69
Liver	GSE17856	GPL6480	25075	13802	43	44	87
Kidney	GSE15641	GPL96	22283	11593	69	23	92
Thyroid	GSE33630	GPL570	54675	12986	60	45	105
Breast	GSE3494	GPL96	22283	12986	60	176	236

**Table 2 genes-14-00574-t002:** Mean AUC of e-DRW.

Restart Probabilities	Classifiers	Datasets					
		Lung	Stomach	Liver	Kidney	Thyroid	Breast
0.1	NB	0.878505	0.846377	0.87931	0.866304	0.939048	0.762712
	KNN	0.918692	0.858066	0.855172	0.838043	0.946667	0.751695
	LR	0.864486	0.795652	0.871264	0.893478	0.875238	0.761017
0.2	NB	0.917757	0.866667	0.872414	0.820652	0.949524	0.75678
	KNN	0.929907	0.915942	0.870115	0.834783	0.917143	0.762288
	LR	0.860748	0.797101	0.851724	0.846739	0.912381	0.758051
0.3	NB	0.966355	0.86087	0.885057	0.802174	0.957143	0.761017
	KNN	0.946729	0.933333	**0.931034**	0.858696	0.954286	0.760169
	LR	0.873832	0.797101	0.918391	0.868478	0.939048	0.761017
0.4	NB	0.935514	0.926087	0.87931	0.873913	**0.96**	0.755932
	KNN	0.948598	0.908696	0.855172	0.883696	0.950476	0.762712
	LR	0.914953	0.842029	0.871264	**0.926087**	0.898095	0.751695
0.5	NB	0.962617	0.905797	0.83908	0.83913	0.950476	0.752542
	KNN	**0.980374**	0.886957	0.855172	0.897826	0.954286	**0.769068**
	LR	0.899065	0.831884	0.837931	0.871739	0.931429	0.755085
0.6	NB	0.969159	0.917391	0.874713	0.863043	0.937143	0.758898
	KNN	0.971028	**0.955072**	0.873563	0.829348	0.868571	0.75678
	LR	0.909346	0.868116	0.862069	0.891304	0.86	0.758475
0.7	NB	0.969159	0.849275	0.868966	0.804348	0.941905	0.761864
	KNN	0.961682	0.83913	0.906897	0.795652	0.94	0.747034
	LR	0.903738	0.792754	0.851724	0.823913	0.900952	0.760593
0.8	NB	0.961682	0.897101	0.918391	0.856522	0.942857	0.754237
	KNN	0.930841	0.892754	0.905747	0.88913	0.921905	0.751271
	LR	0.87757	0.785507	0.827586	0.861957	0.932381	0.755085
0.9	NB	0.931776	0.894203	0.928736	0.86413	0.950476	0.751695
	KNN	0.919626	0.926087	0.910345	0.906522	0.950476	0.747034
	LR	0.909346	0.857971	0.931034	0.894565	0.900952	0.753814

Bold values: the highest values. Refer to Appendix A for more details.

**Table 3 genes-14-00574-t003:** Mean AUC of e-DRW.

Risk Pathways	Datasets					
	Lung	Stomach	Liver	Kidney	Thyroid	Breast
PI3K-Akt signaling pathway	√ *	√	√	√	√	
Pathways in cancer	√	√	√	√	√	
Human papillomavirus infection	√	√	√			
Calcium signaling pathway				√	√	√
ECM-receptor interaction	√				√	
Lipid and atherosclerosis	√					√
Apelin signaling pathway	√			√		
Focal adhesion	√				√	
Hippo signaling pathway		√				√
Wnt signaling pathway		√				√
Small cell lung cancer			√		√	
Neuroactive ligand–receptor interaction				√	√	
Integrin-linked kinase signaling		√		√		
Adrenergic signaling in cardiomyocytes				√		√
cGMP-PKG signaling pathway				√		√

* Detected significant pathways.

## Data Availability

The data analyzed in this paper are available in the Gene Expression Omnibus (GEO) repository at NCBI.

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
