# Peer review of "An Entropy-Based Directed Random Walk for Cancer Classification Using Gene Expression Data Based on Bi-Random Walk on Two Separated Networks"

_genes, 2023, doi:10.3390/genes14030574_

Round 1

Reviewer 1 Report

The work presents an interesting application of random walks to the context of cancer classification based on relevant databases. I believe the notions are properly presented, and and mentions of comparable research is done.

Author Response

Thank you for the suggestions. We have revised and checked our English language in the manuscript.

Reviewer 2 Report

The authors tried to solve the problem of integrating two or more omics data through this study. This reviewer also agrees, and it is expected that it can be solved through the appropriate method suggested in this paper. The method of this study is expected to secure robustness and reproducibility of disease models based on omics data.

1. [Line 23]Directed Random Walk (DRW) -> DRW

2. [Lines 91-92]significant Directed Random Walk (sDRW) -> significant DRW

3. [Lines 67, 70]Please provide full names of PCA and PAC.

4. Why not use the expression Differentially expressed genes (DEGs)? There is confusion because "differential signals of genes [Line 349]" or "differential expression of genes [Line 373]" are used together in one manuscript. Please unify.

5. The third and fourth paragraphs of the Introduction should be explained in the discussion. After moving, please combine the result interpretation and description.

6. [Figure 1]Duplication of "Cleaned gene expression data samples" in Figure 1. Excluding one of the two, please present only one.

7. Table 1 and Table 2 are the same, so please combine them.

8. Methods have unnecessary descriptions, and there are no necessary computer language-based analysis libraries or in-house coding. What computer language is the result coded in? It seems to me that the analysis was based on R. Please present clearly.

9. Continuing to the above point, authors must disclose the source code used for analysis on github or upon request. And, in the "Data Availability Statement" section [Lines 399-400], you must describe how to obtain the source code.

10. The description in [Lines 296-328] is very long. A 10-line description briefly describing Table 2 should suffice. Please consider presenting an abbreviated statement.

11. The second and third paragraphs of the Discussion section both start with "Based on". This makes readers feel bored, so please correct it appropriately.

Author Response

Thank you for the comments and suggestions from reviewer.

Reviewer 3 Report

The paper proposed an entropy-based pathway activity inference scheme for identifying reproducible pathway biomarkers for clinical cancer applications. The literature review is well organized. A lot of datasets are analyzed.

Very good comparison with existing methods.

However, I have a small suggestion to improve the paper:

1) Formula (5) represents classical min-max normalization, and this is the classical score, as well as formulas (6)-(7). Is it necessary to put them on paper?

2) It would be great to investigate whether the datasets are balanced. It means is AUC enough for analysis?

Author Response

Thank you for the comments and suggestions.

Round 2

Reviewer 2 Report

The authors completed the revision by reflecting the opinions of the reviewers.

It is hoped that the presented "eDRW" R package will be made public so that many people can utilize it.

I hope that this paper will be published and the eDRW algorithm will be used for many DEG results.

Reviewer 3 Report

I haven't any comments